 eLife

# Witnessing the structural evolution of an RNA enzyme

Xavier Portillo[1], Yu-Ting Huang[2], Ronald R Breaker[1,3], David P Horning[2]*, Gerald F Joyce[2]*

[1]Department of Molecular, Cellular and Developmental Biology, Yale University, New Haven, United States; [2]The Salk Institute, La Jolla, United States; [3]Howard Hughes Medical Institute, New Haven, United States

**Abstract** An RNA polymerase ribozyme that has been the subject of extensive directed evolution efforts has attained the ability to synthesize complex functional RNAs, including a full-length copy of its own evolutionary ancestor. During the course of evolution, the catalytic core of the ribozyme has undergone a major structural rearrangement, resulting in a novel tertiary structural element that lies in close proximity to the active site. Through a combination of site-directed mutagenesis, structural probing, and deep sequencing analysis, the trajectory of evolution was seen to involve the progressive stabilization of the new structure, which provides the basis for improved catalytic activity of the ribozyme. Multiple paths to the new structure were explored by the evolving population, converging upon a common solution. Tertiary structural remodeling of RNA is known to occur in nature, as evidenced by the phylogenetic analysis of extant organisms, but this type of structural innovation had not previously been observed in an experimental setting. Despite prior speculation that the catalytic core of the ribozyme had become trapped in a narrow local fitness optimum, the evolving population has broken through to a new fitness locale, raising the possibility that further improvement of polymerase activity may be achievable.

*For correspondence:
dhorning@salk.edu (DPH);
gjoyce@salk.edu (GFJ)

Competing interest: The authors declare that no competing interests exist.

## Introduction

Directed evolution in the laboratory has proven to be a powerful means for obtaining proteins and nucleic acids with desired functional properties. Starting from molecules that contain either regions of random sequence or a defined sequence that has been diversified, one carries out iterative rounds of selection and amplification to obtain ever more fit variants in pursuit of the desired phenotype. This process is analogous to Darwinian evolution in biology, except that the fitness criteria are imposed by the experimenter, rather than being the result of natural selection.

Whereas biological evolution has been operating on Earth for billions of years across highly diverse environments, directed evolution experiments typically involve only a dozen 'generations' and are confined to narrowly defined reaction conditions. Thus, it is perhaps not surprising that the trajectory of evolution in the laboratory tends to follow a narrow path. Once a functional motif has been defined, subsequent rounds of evolution typically refine rather than remodel that motif. In some cases, the experimenter intervenes by appending a new region of random or defined sequence that can evolve into a new structural domain (*Jaeger et al., 1999*; *Johnston et al., 2001*; *Ikawa et al., 2004*) or evolves a different function that emerges together with a new structural motif (*Lorsch and Szostak, 1994*; *Schultes and Bartel, 2000*; *Huang and Szostak, 2003*). However, more extensive evolution appears to be required to achieve tertiary structural remodeling while maintaining the same function throughout the evolutionary process.

Here, from the perspective of 52 consecutive rounds of directed evolution under progressively more demanding selection constraints, an RNA polymerase ribozyme was seen to undergo a tertiary structural change, similar to changes that are inferred to have occurred in nature based on phylogenetic

analyses (*Gutell et al., 1994*; *Williams and Bartel, 1996*; *Pfingsten et al., 2007*). Multiple evolutionary pathways were explored by the evolving population of RNA molecules as they transitioned from one structural configuration to another, ultimately converging upon a new fold that results in improved catalytic activity. The details of this transition are witnessed by a combination of structural, biochemical, and deep sequencing analyses, providing a clear-eyed view of the molecular evolution of structural innovation.

The RNA-catalyzed polymerization of RNA has received special attention because it is thought to be the central function of the 'RNA world', a time in the early history of life, prior to the emergence of DNA and proteins, when RNA served as both the genetic material and the chief agent of catalytic function (*Crick, 1968*; *Gilbert, 1986*; *Joyce, 2002*). An RNA enzyme that catalyzes the RNA-templated copying of RNA could, in principle, generate additional copies of itself and thus serve as the basis for self-sustained Darwinian evolution. No such enzyme currently exists, although diligent efforts by several laboratories have used directed evolution to isolate an RNA ligase ribozyme from a population of random-sequence RNAs (*Bartel and Szostak, 1993*; *Ekland et al., 1995*), then drive the ribozyme to function as an ever more efficient RNA polymerase (*Johnston et al., 2001*; *Zaher and Unrau, 2007*; *Wochner et al., 2011*; *Horning and Joyce, 2016*; *Cojocaru and Unrau, 2021*), now with the ability to synthesize RNAs as complex as the parental ligase (*Attwater et al., 2018*; *Tjhung et al., 2020*).

All of the previously described descendants of the original class I ligase ribozyme retain the same catalytic core. In 2001, Bartel and colleagues appended 76 random-sequence nucleotides to the 3' end of the ligase and selected for its ability to catalyze the polymerization of nucleoside 5'-triphosphates (NTPs). This effort resulted in a novel 'accessory domain' that enables the addition of up to 14 successive NTPs on the most favorable templates (*Johnston et al., 2001*). Following further evolutionary optimization, the maximum length of extension by the polymerase was increased to 20 NTPs (*Zaher and Unrau, 2007*). Holliger and colleagues then added 48 random-sequence nucleotides to the 5' end of the ribozyme and selected for its ability to catalyze multiple NTP additions. This procedure resulted in the discovery of a 'processivity tag' that forms a region of Watson-Crick pairing between the 5' end of the ribozyme and the 5' end of the template, enabling addition of up to 95 NTPs on a template that contains multiple repeats of an especially favorable sequence (*Wochner et al., 2011*). Throughout these many rounds of evolution, only a single point mutation became fixed within the core ligase domain, converting a G-C pair to a G-U wobble pair. However, the combination of the added accessory domain and processivity tag, hereafter referred to as the 'wild type', provided a more robust polymerase that made it possible to impose more stringent selection criteria going forward.

In two subsequent studies, the wild-type polymerase ribozyme was further evolved by requiring it to synthesize functional RNAs, with selection of the ribozyme being dependent on the function of the synthesized product. In the first study, the ribozyme was required to synthesize two different RNA aptamers, each involving the copying of a challenging template (*Horning and Joyce, 2016*). The resulting '24-3' polymerase, obtained after 24 rounds of evolution, has substantially improved activity compared to its predecessors, especially when copying structured templates with heterogeneous base composition. It is able to synthesize the entire 33-nucleotide hammerhead ribozyme, which became the requirement for selection in the second study. Another 14 rounds of evolution were then carried out, culminating in the '38-6' polymerase, which is ~10 -fold more active than the 24-3 polymerase and can more efficiently synthesize complex RNA products, such as yeast phenylalanyl-tRNA (*Tjhung et al., 2020*).

This lineage continues in the present study, which began with a population of variants of the 38-6 polymerase that had been randomized at a frequency of 10 % per nucleotide position, and entailed 14 additional rounds of evolution that sought to improve both the activity and fidelity of the polymerase. The resulting '52-2' polymerase is indeed further improved, but also reveals that the ribozyme underwent structural rearrangement of its catalytic core, enabled by 11 substitution, 2 insertion, and 2 deletion mutations that accumulated over the course of evolution starting from the wild-type polymerase. An existing stem element became shortened while a new stem element was formed, together creating a pseudoknot structure that lies in close proximity to the ribozyme's active site. This new structure became stabilized over time through the sampling and fixation of successive mutations, providing a compelling demonstration of the blind inventiveness of Darwinian evolution. The new

structure also shows that the ribozyme was not trapped on a local fitness peak, but instead is actively evolving, with the opportunity to explore novel regions of sequence space.

# Results

## Advanced evolution of an RNA polymerase ribozyme

The 38-6 polymerase ribozyme contains 182 nucleotides, with 20 substitution, 4 insertion, and 2 deletion mutations compared to the wild-type ribozyme (*Tjhung et al., 2020*). Random mutations were introduced throughout the 38-6 polymerase at a frequency of 10 % per nucleotide position, excluding 14 nucleotides at the 5' end and 15 nucleotides at the 3' end that served as primer binding sites for amplification of the selected RNAs. A starting population of approximately three copies each of 4 × $10^{14}$ different RNAs was used to initiate subsequent rounds of directed evolution, requiring the polymerase to synthesize a functional hammerhead ribozyme and further requiring the polymerase to operate under conditions of reduced $Mg^{2+}$ ion concentration (*Supplementary file 1*). The latter constraint sought to provide conditions that may be conducive to increased polymerase fidelity (*Eckert and Kunkel, 1990*; *Achuthan et al., 2014*) and to the reduced degradation of RNA. Fourteen rounds of evolution were carried out, performing error-prone PCR during most rounds to maintain genetic diversity in the population, although not during the final two rounds so that the population could converge on the fittest variants.

Following the 14th round, 52 rounds in total relative to the wild-type polymerase, 30 individuals were cloned from the population and sequenced. The majority of these individuals, including the 52-2 polymerase that dominated the final population, exhibited increased activity compared to the 38-6 polymerase in the presence of either standard (200 mM) or reduced (50 mM) concentrations of $Mg^{2+}$. A dominant clone, termed '52-2', was chosen for further study. It contains four mutations relative to the 38-6 polymerase, is 3-fold more efficient in synthesizing the hammerhead ribozyme (*Figure 1A*), and is 23-fold more efficient in synthesizing the class I ligase (*Figure 1B*). The synthesized ligase is catalytically active, with an observed rate of RNA-templated RNA ligation of 0.31 ± 0.02 $hr^{-1}$, which corresponds to a rate acceleration of 1500-fold compared to the uncatalyzed reaction (*Figure 1C*). However, this rate is substantially lower than that of the class I ligase synthesized by T7 RNA polymerase, which has an observed rate of 6.3 ± 0.6 $min^{-1}$ under the same reaction conditions.

Based on the modest fidelity of the 24-3 and 38-6 polymerases (*Tjhung et al., 2020*), it is likely that ligase molecules synthesized by the 52-2 polymerase contain multiple mutations, which may reduce or eliminate catalytic activity. Deep sequencing was carried out to analyze both the hammerhead and class I ligase ribozymes synthesized by the 52-2 polymerase. For the hammerhead, synthesized in the presence of either 200 or 50 mM $Mg^{2+}$, the average fidelity per nucleotide position was 91.7 % or 94.4%, respectively (*Supplementary file 2*). The most common mutations are the result of G•U wobble pairing, with all types of mutations being less frequent in the presence of the lower concentration of $Mg^{2+}$. For the class I ligase, synthesized in the presence of 200 mM $Mg^{2+}$, the average fidelity was 84.1 %. There are an average of 12 mutations per copy of the ligase, which explains its reduced activity compared to that of the protein-synthesized material.

## Sequence changes over the course of evolution

The 24-3, 38-6, and 52-2 polymerases represent distinct points along a lineage that has diverged substantially from the wild type. Overall, the 52-2 polymerase differs by 26 mutations compared to the wild type, which corresponds to 14 % of its total sequence (*Figure 2—figure supplement 1*). Fifteen of these mutations are within the catalytic core. Two core mutations that arose between the 38-6 and 52-2 polymerase are notable because they change A and U residues at positions 15 and 85 to C and G, respectively. Position 15 is the first nucleotide beyond the fixed primer binding site and had been thought to be part of a single-stranded region (termed J1/3) that helps to position a catalytic $Mg^{2+}$ ion within the active site of the parental ligase ribozyme (*Shechner et al., 2009*; *Shechner and Bartel, 2011*). Position 85 lies within what was thought to be a loop region that closes the P7 stem of the catalytic core, but otherwise has no functional importance (*Ekland and Bartel, 1995*). Yet, these two mutations arising in concert raised suspicion that they might form a Watson-Crick pair, which would require a very different structural arrangement within the catalytic core.

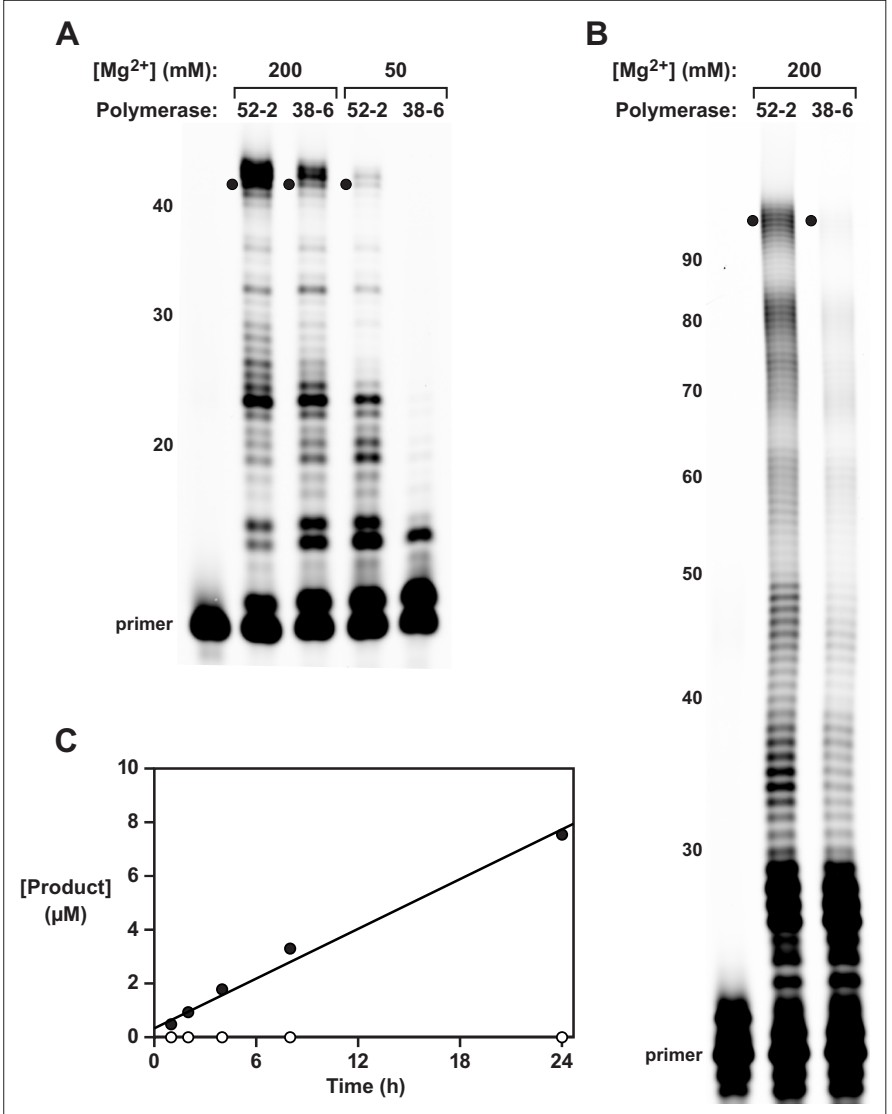

**Figure 1.** Synthesis of functional RNA molecules by the 38-6 and 52-2 polymerases. (**A**) Synthesis of the hammerhead ribozyme in the presence of either 50 or 200 mM $Mg^{2+}$ after 1 hr. (**B**) Synthesis of the class I ligase ribozyme in the presence of 200 mM $Mg^{2+}$ after 24 hr. Reaction conditions for (**A**) and (**B**): 100 nM polymerase, 80 nM primer, 100 nM template, 4 mM each NTP, and either 50 or 200 mM $MgCl_2$ at pH 8.3 and 17 °C. Intermediate-length products are numbered at the left. Black dots indicate full-length products. (**C**) Time course of RNA ligation catalyzed by the class I ligase ribozyme that had been synthesized by the 52-2 polymerase, as shown in (**B**), comparing the RNA-catalyzed (black circles) and the uncatalyzed (white circles) reactions, which have a rate of 0.31 and 0.00021 $hr^{-1}$, respectively. Reaction conditions: ±1 µM ligase ribozyme, 20 µM 5'-substrate, 80 µM 3'-substrate, 60 mM $MgCl_2$, 200 mM KCl, and 0.6 mM EDTA at pH 8.3 and 23 °C.

The online version of this article includes the following figure supplement(s) for figure 1:

**Source data 1.** Gel images (raw and annotated) of hammerhead (*Figure 1A*) and class I ligase (*Figure 1B*) ribozymes synthesized by the 52-2 polymerase.

**Source data 2.** Product yields and linear regression parameters for the reaction catalyzed by the class I ligase that had been synthesized by the 52-2 polymerase (*Figure 1C*).

Neither the J1/3 nor the P7 portion of the ribozyme was mutated during the many rounds of evolution leading from the ligase to the wild-type polymerase. However, both of these regions accumulated numerous mutations during the subsequent rounds of evolution, with seven changes in the 24-3 polymerase, six more in the 38-6 polymerase, and two more in the 52-2 polymerase (*Figure 2B*). Taken together, these mutations suggest that a new stem, termed P8, has evolved. One strand of

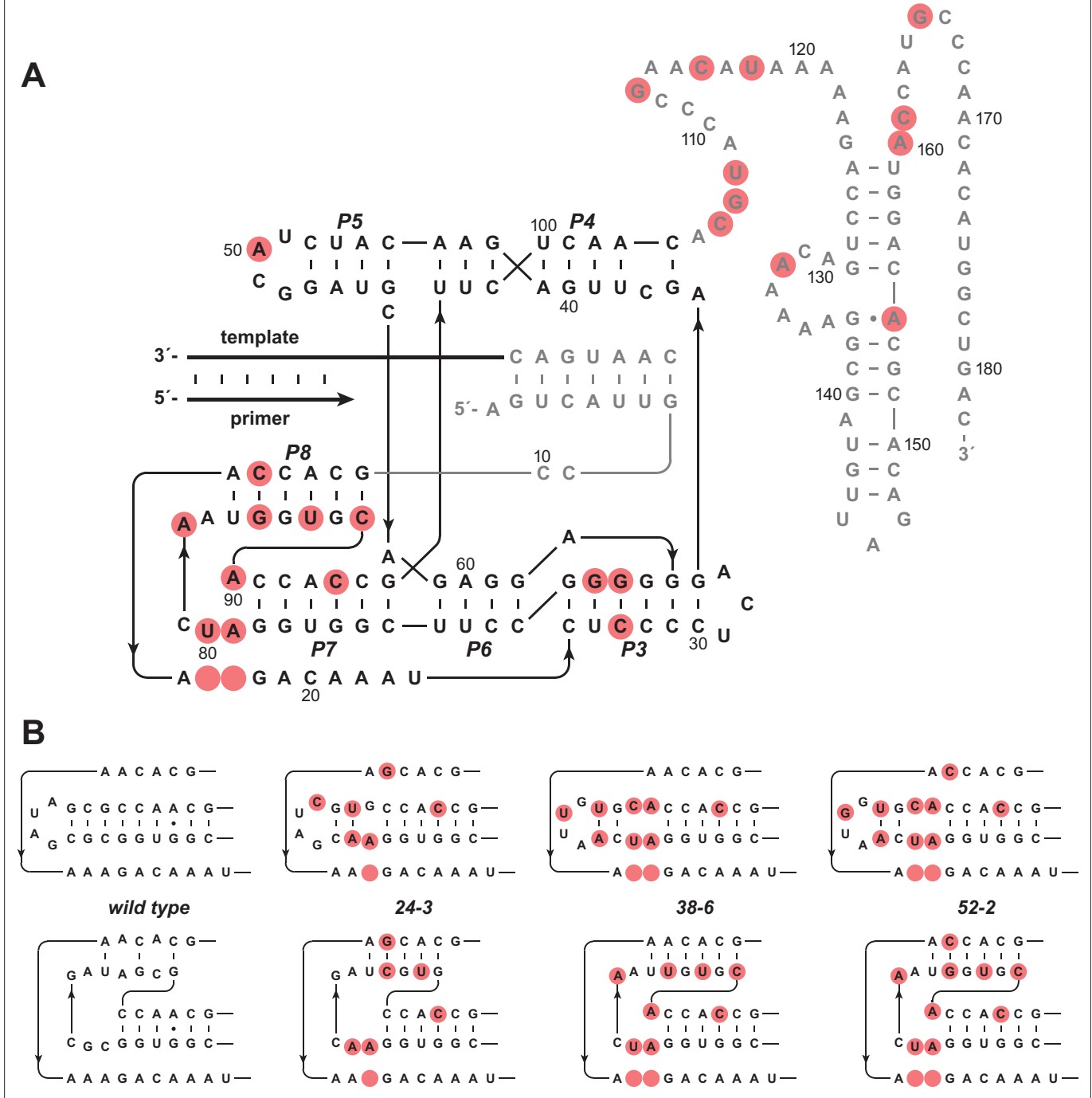

**Figure 2.** Evolution of the novel pseudoknot structure. (**A**) Sequence and secondary structure of the 52-2 polymerase. Red circles indicate mutations relative to the wild-type polymerase. Regions outside the ligase core are shown in gray. Paired regions and nucleotides are numbered according to the 52-2 polymerase, with corresponding nucleotides numbered similarly for all polymerase variants. (**B**) Progressive mutation of the region encompassing the P7 and P8 stems, mapped onto both the old structure (*top*) and the new pseudoknot structure (*bottom*). Blank circles indicate deletions.

The online version of this article includes the following figure supplement(s) for figure 2:

**Figure supplement 1.** Aligned sequences of the named polymerase ribozymes.

**Figure supplement 2.** Installing the pseudoknot structure on the wild-type background.

**Figure supplement 2—source data 1.** Gel images (raw and annotated) of the reaction catalyzed by the 52-2, wild-type, and chimeric polymerases (*Figure 2—figure supplement 2C*).

this stem contains six nucleotides derived from the J1/3 region, while the complementary strand contains six nucleotides derived from the P7 stem-loop. The proximal portion of the P7 stem appears to remain intact. The new P8 stem would result in a pseudoknot structure that alters the orientation of the P7-derived nucleotides in a manner that is mutually exclusive to the prior core structure of the ribozyme.

## Mutagenesis studies in support of the novel structure

Site-directed mutagenesis studies were carried out to investigate the hypothesized structural rearrangement of the catalytic core. Putative base pairs within the P7 and P8 stems were mutated to identify disruptive and compensatory mutations that may be indicative of Watson-Crick pairing. Each of the six base pairs within the proposed P8 stem was mutated on each of the two strands, together with their combined mutation that would restore complementarity. All 18 of these constructs were evaluated for polymerase activity using a moderately challenging template that required synthesis of the sequence 5'-GUGUGGAGUGACCUCUCCUGUGUGAGUG-3'. On this template, the 52-2 polymerase extends a primer to form full-length products in 20 % yield after 30 min. Each of the single mutations reduced this activity by at least 50-fold, whereas most of the corresponding double mutations restored activity (*Figure 3A*).

For three central pairs of the stem (C12-G88, A13-U87, and C14-G86), activity was nearly fully restored in the corresponding double mutant (G12-C88, U13-A87, and G14-C86). For the two adjacent pairs (G11-C89 and C15-G85), activity was substantially increased in the corresponding double mutant (C11-G89 and G15-C85), although not fully to the level of the 52-2 polymerase. For the most distal pair (A16-U84), mutation of A16 was highly disruptive and activity could not be restored through compensatory mutation. In the class I ligase, the nucleotide corresponding to A16 is known to make a base-specific contact with the template-primer duplex (*Shechner and Bartel, 2011*), and therefore may be required to play a similar role in the polymerase. Nonetheless, sequence covariation in support of this distal pairing is observed among other evolved variants (e.g., G16-C84 and C16-G84), perhaps requiring accompanying mutations to compensate for substitution of A16.

These data supporting the existence of the P8 stem need to be reconciled with the consequences for the P7 stem. Nucleotides 84–89 were previously required to form the P7 stem-loop, with U87, G88, and C89 engaging in Watson-Crick pairs to close one end of the stem. However, mutations that would be expected to disrupt the pairing of either U87-A82 or G88-C81 did not have a deleterious effect, nor was there a beneficial effect of mutations that would be expected to provide additional pairing of C89-G80 (*Figure 3—figure supplement 1*). Thus, the distal portion of the P7 stem-loop no longer appears to form, with some of those nucleotides instead helping to form the new P8 stem.

Kinetic studies were carried out to assess more quantitatively the effect of disruptive and compensatory mutations within the P8 stem. The 52-2 polymerase was compared to the C12G and G88C single mutants, as well as the corresponding double mutant, in a reaction involving an 11-nucleotide templating region that enables measurement of both the rate of the first NTP addition and the average rate of NTP additions across the entire template sequence. The full-length extension product of this reaction has the sequence 5'-UGCGAAGCGUG-3'.

The reaction with the 52-2 polymerase exhibits first-order kinetics, with a $k_{obs}$ of 0.031 min$^{-1}$. However, there is a substantial burst phase, with 18 % of the template-bound primers extended within the first 10 s of the reaction (*Figure 3B*). Multiple nucleotide additions are seen during this short burst phase, including full-length products, suggesting that there is a subpopulation of molecules with a very rapid rate of reaction. The average rate of NTP addition across the entire template during the first 30 s of the reaction is 3.1 min$^{-1}$, which is the fastest rate measured for a polymerase ribozyme (*Figure 3C*). The C12G and G88C mutant polymerases each have substantially lower activity, with a $k_{obs}$ of 0.0039 and 0.0056 min$^{-1}$, and an average rate of NTP addition during the initial phase of the reaction of 0.033 and 0.45 min$^{-1}$, respectively. The amplitude of the burst phase and the rate of the first NTP addition are also substantially lower for the two single mutants. For the compensatory double mutant, however, all of these rates are restored to nearly that of the 52-2 polymerase, with a $k_{obs}$ of 0.030 min$^{-1}$, burst-phase amplitude of 17%, and average rate of NTP addition during the first 30 s of the reaction of 2.3 min$^{-1}$.

The 52-2 polymerase was tested with substantially longer templates to determine the extent to which NTP addition can continue in the burst phase. These templates encoded either 5 or 10 repeats

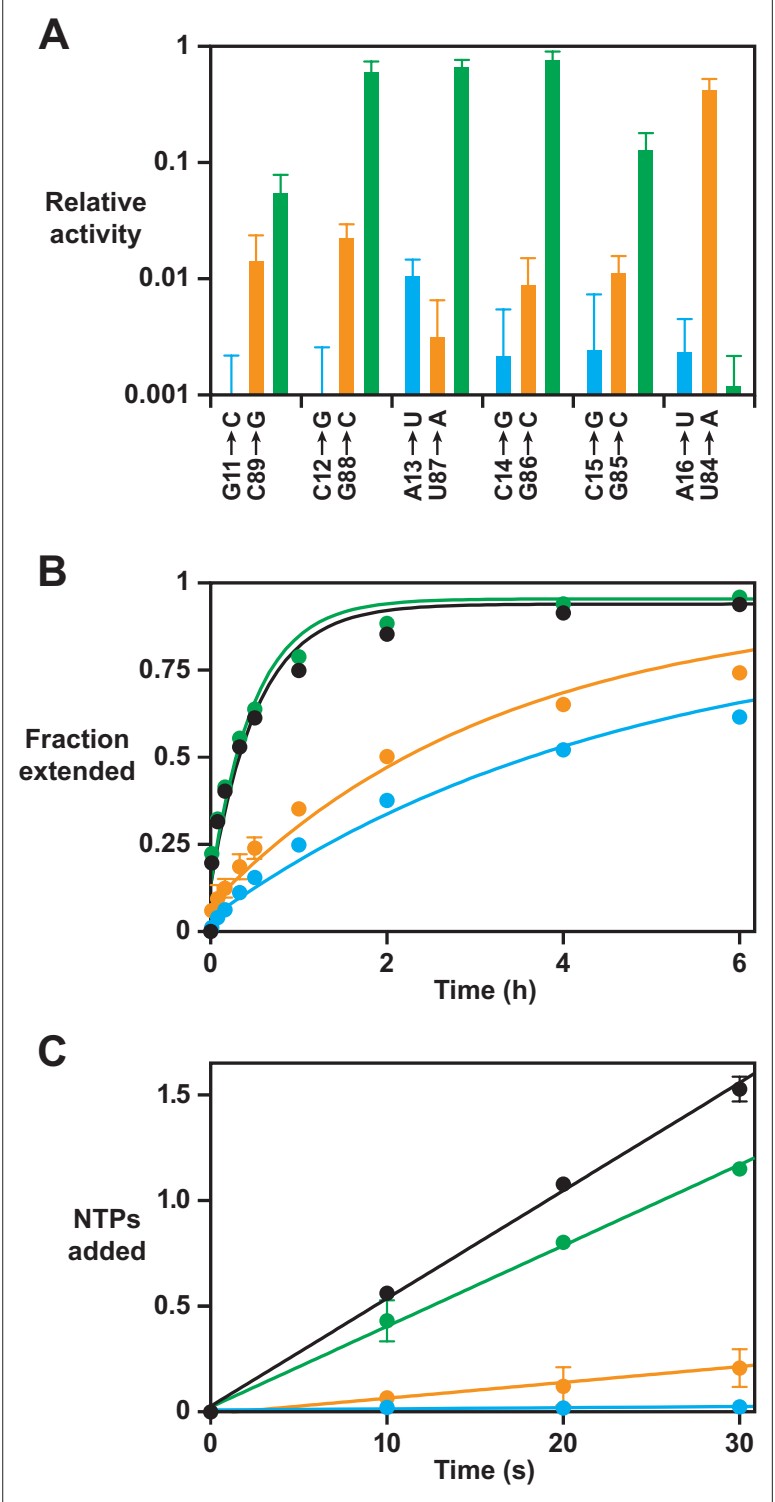

**Figure 3.** Effect on polymerase activity of disruptive and compensatory mutations within the P8 stem. (**A**) Yield of full-length RNA relative to that of the 52-2 polymerase in a 30 min reaction requiring the addition of 28 nucleoside 5'-triphosphates (NTPs) (note the logarithmic scale). At each position within the P8 stem, a transversion mutation was made in either the 5' strand (blue) or the 3' strand (gold), or the two mutations were combined to restore complementarity (green). Values are the average of at least three replicates with standard deviation. (**B**) Time course of primer extension by at least one nucleotide on an 11-nucleotide template, comparing the 52-2 polymerase (black), C12G mutant (blue), G88C mutant (gold), and C12G/G88C double mutant (green). The

*Figure 3 continued on next page*

*Figure 3 continued*

data were fit to a single exponential rise to maximum, allowing for an initial burst phase. (**C**) Average number of nucleotides added during the first 30 s of the reactions depicted in (**B**). The data were fit to a linear equation. For both (**B**) and (**C**), values are the average of two replicates with standard deviation. Reaction conditions: 100 nM polymerase, 100 nM template, 80 nM primer, 4 mM each NTP, and 200 mM $MgCl_2$ at pH 8.3 and 17 °C.

The online version of this article includes the following figure supplement(s) for figure 3:

**Source data 1.** Product yields and regression parameters for the reaction catalyzed by the standard and various mutant forms of the 52-2 polymerase (*Figure 3* and *Figure 3—figure supplement 1*).

**Figure supplement 1.** Effect on polymerase activity of mutations within the P7 and P8 stems.

**Figure supplement 2.** Burst-phase synthesis on long RNA templates.

**Figure supplement 2—source data 1.** Gel images (raw and annotated) of the burst-phase reaction by the 52-2 polymerase on long RNA templates (*Figure 3—figure supplement 2*).

of the sequence 5′-UGCGAAGCGUG-3′, which is known to be especially favorable for synthesis by the wild-type polymerase (*Wochner et al., 2011*). Again with a burst amplitude of ~20%, the burst phase was found to continue, with detectable full-length products after 5 min for the template with 5 repeats and after 10 min for the template with 10 repeats (*Figure 3—figure supplement 2*). By 20 min, the yield of full-length products was 12.9% and 2.3%, respectively.

The mutagenesis studies suggest that the new pseudoknot structure is necessary, but not whether it is sufficient, for the improved catalytic activity of the 52-2 polymerase, which contains 15 additional mutations outside the region of the pseudoknot (*Figure 2A*). A chimeric molecule was constructed by 'transplanting' the pseudoknot onto the wild-type polymerase background, but otherwise maintaining the wild-type sequence (*Figure 2—figure supplement 2A,B*). This chimeric molecule has comparable activity to that of the 52-2 polymerase, including the initial burst phase behavior for NTP addition, which is not seen for the wild type (*Figure 2—figure supplement 2C*). It is notable that for the chimeric molecule, and especially for the 52-2 polymerase, primer extension continues a few nucleotides beyond the templating region and into the oligoadenylate spacer that links the templating region to the processivity tag.

## Structural probing of the wild-type and evolved polymerases

The secondary structure of the wild-type, 24-3, 38-6, and 52-2 polymerases was mapped by in-line probing, a technique that measures the susceptibility of each phosphodiester linkage to spontaneous cleavage (*Soukup and Breaker, 1999*; *Regulski and Breaker, 2008*). Unstructured single-stranded regions of RNA, such as loops and linkers, are more susceptible to spontaneous cleavage because their greater backbone flexibility allows the ribose 2′-hydroxyl to access an in-line geometry with regard to the adjacent phosphate, as is required for the cleavage event.

Comparison of the wild-type and 52-2 polymerases revealed that the latter is less susceptible to cleavage at nucleotide positions 11–14, which correspond to one of the two strands of the P8 stem (*Figure 4A*). Nucleotides 86–88, which correspond to the other strand of P8, are protected in both polymerases, although these nucleotides would be part of the P7 stem in the wild-type polymerase. Conversely, nucleotides 79–81 are more susceptible to cleavage in the 52-2 polymerase, these nucleotides no longer being part of the P7 stem. The retained portions of the P7 stem, nucleotides 73–78 and 91–96, are well protected from cleavage in both the wild-type and 52-2 polymerases. Note that there is a two-nucleotide insertion in the 52-2 polymerase at positions 89–90, which lies between the 3′ end of the P8 stem and 5′ end of the P7 stem, and there is strong cleavage at the unpaired nucleotide A90 in the 52-2 polymerase. All of these data are consistent with a rearrangement of the catalytic core that results in formation of a novel pseudoknot structure involving the P8 stem.

In-line probing of the 24-3 and 38-6 polymerases showed that these ribozymes have intermediate structural features relative to the wild-type and 52-2 polymerases (*Figure 4—figure supplement 1*). For all four polymerases, the degree of spontaneous cleavage was measured for each nucleotide in the P7 and P8 regions and mapped onto both the original and evolved structures (*Figure 4B*). In the 24-3 polymerase, there is reduced susceptibility to cleavage at nucleotide positions 11–14 and enhanced cleavage at positions 79–81, both of which are more pronounced in the 38-6 polymerase. The two-nucleotide insertion first appears in the 38-6 polymerase, with some susceptibility to cleavage

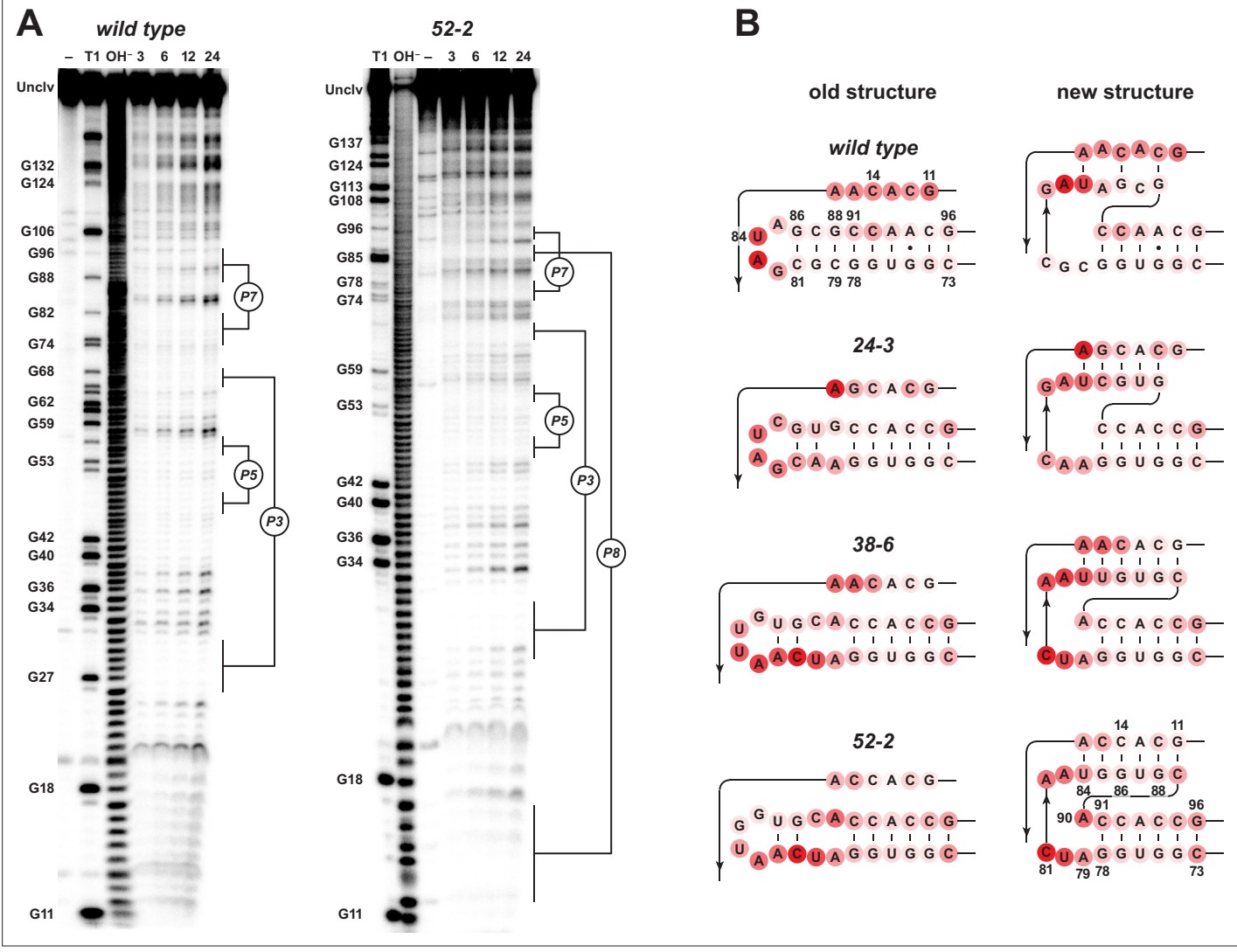

**Figure 4.** Analysis of polymerase structure by in-line probing. (**A**) Polyacrylamide gel electrophoresis analysis of [5'-³²P]-labeled wild-type and 52-2 polymerases, incubated under polymerization conditions for 3, 6, 12, or 24 hr, in comparison to unincubated material (–) and material that had been subjected to partial digestion with either RNase T1 (cleaves after G residues) or NaOH. Full-length polymerase (unclv) and various G residues are labeled at the left; stem regions are labeled at the right. (**B**) Sensitivity to in-line cleavage mapped onto the P7 and P8 stems for both the old structure (*left*) and the new pseudoknot structure (*right*). For each polymerase, red circles of varying intensity indicate % cleavage after 24 hr at each nucleotide position relative to the position with the highest level of cleavage within the region shown.

The online version of this article includes the following figure supplement(s) for figure 4:

**Source data 1.** Gel images (raw and annotated) of in-line probing of the wild-type and 52-2 polymerases (*Figure 4A*).

**Source data 2.** Quantitation of in-line probing data at positions 11–16 and 83–96 of the wild-type, 24-3, 38-6, and 52-2 polymerases after 24 hr (*Figure 4B*).

**Figure supplement 1.** Analysis of polymerase structure by in-line probing.

**Figure supplement 1—source data 1.** Gel images (raw and annotated) of in-line probing of the 24-3 and 38-6 polymerases.

at position 90, and this susceptibility becomes more pronounced in the 52-2 polymerase. The retained portion of the P7 stem, nucleotides 73–78 and 91–96, is well protected from spontaneous cleavage for all of the ribozymes in the evolutionary lineage.

## Sequence variation over the course of evolution

Deep sequencing analysis was carried out to investigate the population dynamics that underlie the emergence of the novel structure. Sequences were obtained after rounds 6, 8, 11, 12, 14, 16, 18, 21,

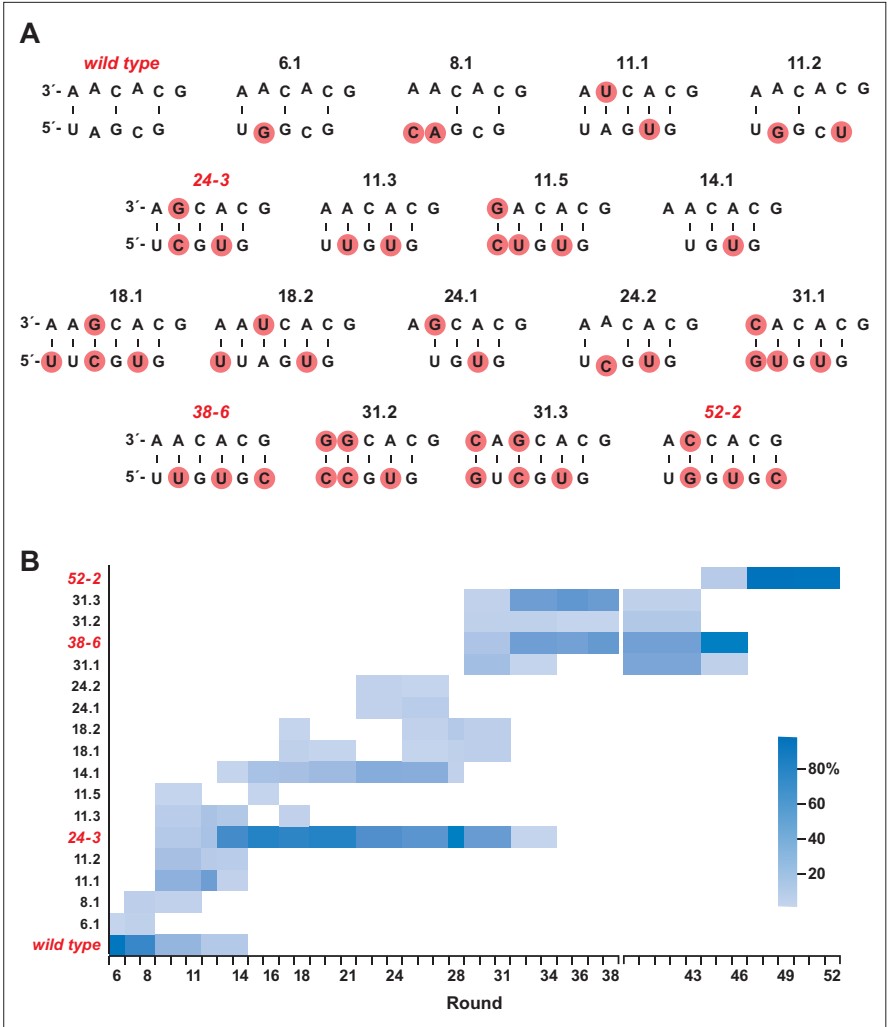

**Figure 5.** Composition of the evolving population. (**A**) The 18 most highly represented sequence clusters for the P8 stem. Clusters that include the wild-type, 24-3, 38-6, and 52-2 polymerases are named for those polymerases; all other clusters are named according to the round in which they first appeared at a frequency of >1%. Red circles indicate mutations relative to the wild type. Note that for clusters 14.1 and 24.1, an A residue has been deleted in the bottom strand. (**B**) Heatmap depicting the representation of the 18 clusters over the course of evolution (scale bar at right). Axis break after round 38 indicates that the 38-6 polymerase was isolated from the population and mutated at a frequency of 10 % per nucleotide position before resuming directed evolution.

The online version of this article includes the following figure supplement(s) for figure 5:

**Figure supplement 1.** Composition of each strand of the P8 stem over the course of evolution.

**Figure supplement 1—source data 1.** Frequency of occurrence of variants of the 5′ and 3′ strands of the P8 stem over the course of evolution.

24, 27, 28, 31, 34, 36, 38, 43, 46, 49, and 52. These sequences were aligned and clustered based on the region encompassing the P8 stem (**Figure 5**; **Supplementary file 3**). Clusters representing >1% of the population in any given round were identified and those that were present in only a single round at <5 % frequency were ignored. This analysis resulted in a total of 105 clusters, representing ~95 % of all sequence reads. Looking across all 52 rounds, there are 18 highly represented clusters that differ with regard to the sequence of the P8 stem, which include the wild-type, 24-3, 38-6, and 52-2 polymerases. These 18 variants are shown in **Figure 5**, together with their frequency of occurrence over time.

During the early rounds of evolution, the wild-type sequence continued to dominate, but became extinct after round 14. Other clusters that appeared early and lacked complementarity in the region

of the P8 stem also became extinct after round 14. Most of the clusters that arose subsequently over the course of evolution had five Watson-Crick pairs in the region of the P8 stem. The sequences of the 24-3, 38-6, and 52-2 polymerases first became apparent at rounds 11, 31, and 46, respectively, and were the dominant cluster at the time they were first isolated from the population, following rounds 24, 38, and 52, respectively.

Among the distinct forms of the P8 stem that became abundant over the course of evolution, most contained five base pairs, but some contained either four or six base pairs. All of the variability among these sequences occurs at the distal end of the P8 stem, whereas the proximal nucleotides G11, C12, A13, and C14 (and their pairing partners) are universally conserved. This is not surprising because nucleotides 11–14 are part of the primer binding site that remains fixed during each round of selective amplification. Nucleotides 15 and 16 are free to vary, and do so, so long as complementarity is maintained with the corresponding nucleotides of the opposing strand.

The 5′ and 3′ strands of the P8 stem were also considered individually to determine whether the frequency of occurrence of particular variants of the two strands is correlated over time. Because sequence variation within the 5′ strand is limited to two nucleotides, there was insufficient variation to track all 18 highly represented clusters, but this could be done for variants corresponding to the wild-type, 24-3, 38-6, and 52-2 polymerases (*Figure 5—figure supplement 1*). For each of the major polymerase species, there is a high degree of correlation for the occurrence of paired variants of the 5′ and 3′ strands, indicating that they rose and fell together.

The sequence of the P8 stem that occurs in the 52-2 polymerase first emerged at round 46, and by round 49 constituted 98 % of the population. During that same interval, the 38-6 form of the polymerase fell to extinction. Although unlikely to be the last chapter in the evolution of the polymerase ribozyme, the 52-2 polymerase strongly consolidates the pseudoknot structure within the catalytic core, providing robust activity for the RNA-catalyzed synthesis of complex RNAs.

## Discussion

The class I ligase ribozyme was evolved from a starting pool of random-sequence RNAs nearly 30 years ago (*Bartel and Szostak, 1993*), and has been subjected to more rounds of directed evolution, as either a ligase or the core component of a polymerase, than any other ribozyme. It has proven to be a remarkably stable motif, perhaps due to the high catalytic efficiency of the parental ligase, which is comparable to that of RNA ligase proteins and approaches the physical limit of substrate recognition through Watson-Crick base pairing (*Bergman et al., 2000*). As a polymerase, however, the ribozyme has considerably lower activity due to its poor affinity for the primer-template complex (*Lawrence and Bartel, 2003*). This limitation has been partially circumvented by adding a processivity tag that enables the ribozyme to bind tightly to the template though Watson-Crick pairing (*Wochner et al., 2011*). More recently, the polymerase was evolved to recognize a 'promoter' sequence on the template through a clamp-like mechanism, which enables it to operate in a more processive manner (*Cojocaru and Unrau, 2021*). Extensive rounds of directed evolution have also been used to increase the catalytic efficiency and sequence generality of the polymerase, to the level that it is now capable of synthesizing the class I ligase and other complex functional RNAs (*Tjhung et al., 2020*).

It has been suggested that the ribozyme occupies a high and isolated fitness peak, whereby its structural elements are so tightly interwoven that any exploration of alternative structures would have severe negative consequences for fitness (*Ellington, 2008*). Similar isolation in sequence space has been observed for smaller artificial ribozymes (*Pitt and Ferré-D'Amaré, 2010*; *Blanco et al., 2019*), although it could be argued that the more complex structure of the ligase affords more degenerate tertiary interactions that support greater evolvability compared to simpler structures (*Edelman and Gally, 2001*).

The new pseudoknot structure emerged spontaneously during the directed evolution process and clearly contributes to the improved fitness of the polymerase ribozyme, demonstrating that the ligase core can indeed access alternative structures in response to stringent selection pressure. Notably, the only major structural changes that this motif had undergone previously were also the result of selection pressures that aimed to improve polymerase activity, although those new structures arose from regions of random-sequence nucleotides that were appended to the ends of the motif (*Johnston et al., 2001*; *Wochner et al., 2011*; *Cojocaru and Unrau, 2021*).

Pseudoknots are compact and informationally economical structures that need not alter the global architecture of an RNA (*Gutell et al., 1994*). Thus a pseudoknot was able to evolve within the catalytic core of the polymerase ribozyme without significantly perturbing other structural features. The more processive polymerase variant that was evolved by *Cojocaru and Unrau, 2021*, does not contain, nor could it accommodate, the pseudoknot structure, demonstrating that there are alternative solutions to achieve improved catalytic activity.

Both structural probing and sequence analysis of the evolving population over the course of 52 rounds revealed that the new core structure did not appear suddenly as a 'hopeful monster' (*Goldschmidt, 1940*; *Gould, 1977*), but rather was the result of gradual remodeling of the core through a succession of variants along multiple mutational pathways. Prior studies have shown that the nucleotides that gave rise to the P8 stem are relatively tolerant of mutation (*Petrie and Joyce, 2014*). However, the structural stasis of the ligase fold was broken only when selection required a very challenging enzymatic activity, involving the accurate copying of 10–30 nucleotides from structured RNA templates. Presumably, this phenotype could not have been achieved by more subtle modification of the prior structure. Comparing the core sequence of the 52-2 polymerase to that of the wild type, there are only four substitution and two deletion mutations outside the region of the P7 and P8 stems (*Figure 2*), none of which would alter the secondary and presumed tertiary structure of the ribozyme.

Due to the requirement to provide a primer binding site for selective amplification, the 14 nucleotides at the 5' end of the polymerase were immutable throughout the evolution process. The last four of these nucleotides became part of the new P8 stem, which necessitated a C-to-U mutation at position 87 and a C insertion at position 89 to achieve complementary pairing with the fixed nucleotides. It is tempting to wonder how the evolutionary solution might have been different, perhaps better, if this constraint had not been in place. Clearly, there is sequence flexibility within the region that connects the 3' end of the processivity tag to the 5' end of the P8 stem, suggesting that the nucleotides both upstream and within the 5' half of the P8 stem should be allowed to vary in future rounds of evolution. The new topology of the catalytic core also suggests locations where the insertion of random-sequence nucleotides would be tolerated and may provide an opportunity for further evolutionary improvement.

The new pseudoknot structure results in more than a change of primary and secondary structure, also having remodeled the tertiary structure of the catalytic core. One strand of the P8 stem derives from the former J1/3 region and the other from the distal portion of the P7 stem-loop. Based on the X-ray crystal structure of the class I ligase, the P7 stem-loop had previously been oriented away from the active site (*Shechner et al., 2009*), but the new topology draws those nucleotides back toward the active site. The crystal structure also shows that the J1/3 region lies in direct contact with the minor groove of the primer-template duplex, with three adenosine residues of J1/3 forming A-minor interactions with nucleotides located 3 and 4 positions upstream of the ligation junction. One of those adenosines has been deleted in the 52-2 polymerase. Other residues of J1/3, which are retained in the 52-2 polymerase, coordinate a $Mg^{2+}$ ion that helps to catalyze the phosphoester transfer reaction (*Shechner and Bartel, 2011*).

The comparative in-line probing studies show that there has been subtle alteration of the polymerase structure in regions beyond the P7 and P8 stems (*Figure 4—figure supplement 1*), most notably in the P3 and P6 stems that form a coaxial stack with P7 (*Shechner et al., 2009*). There also were changes in peripheral regions of the ribozyme, perhaps secondary adaptations to the changes that occurred within the catalytic core. The burst-phase kinetics of the 52-2 polymerase suggests that it can adopt multiple folded states, some that are highly active and might be stabilized by further evolution through a combination of core and peripheral mutations.

Further improvement of the polymerase, especially with regard to template processivity and copying fidelity, will be required to develop a general RNA replicase. Fidelity is a major obstacle if the aim is to synthesize functional products as long as the polymerase itself. A previous study demonstrated that there is a trade-off between product length and fidelity, especially when copying challenging templates (*Tjhung et al., 2020*). The more time that is required to complete the synthesis, the more opportunity there is to extend a mismatched terminus and thereby incorporate a mutation among the full-length materials. In synthesizing the hammerhead ribozyme in the presence of 200 mM $Mg^{2+}$, the 24-3, 38-6, and 52-2 polymerases all have a fidelity of ~92 % per nucleotide position. The 52-2 polymerase is able to operate in the presence of 50 mM $Mg^{2+}$, and under that condition

the fidelity of hammerhead synthesis improves to 94.4 %. But when synthesizing the class I ligase, which requires 200 mM $Mg^{2+}$ to achieve good yield, the fidelity of the 52-2 polymerase is only 84.1 %. The evolutionary path to substantially improved polymerase fidelity likely will entail both improved catalytic activity and an ability to operate under conditions that are less conducive to base mismatch.

With the heritage of 52 successive generations, it has been illuminating to follow the trajectory of evolution as the population sifted through an astronomical number of possibilities to find those that confer selective advantage. Directed evolution is a highly reductionistic process compared to biological evolution, but has few unseen variables and can provide a detailed picture of how novel sequence begets novel structure and corresponding novel function. The class I ligase motif is old by the standard of ribozymes evolved in the laboratory, but vastly younger than ribozymes found in nature, and thus has not been shaped by long-term selection for evolvability. Nonetheless, through sustained selection for novel function, structural novelty emerged as the population escaped the prior fitness peak and entered a new and more promising fitness regime.

## Materials and methods
### Materials
All oligonucleotides used in this study are listed in *Supplementary file 4*. Synthetic oligonucleotides were either purchased from IDT (Coralville, IA) or prepared by solid-phase synthesis using an Expedite 8909 DNA/RNA synthesizer, with reagents and phosphoramidites from either Chemgenes (Wilmington, MA) or Glen Research (Sterling, VA). RNA templates were prepared by in vitro transcription of synthetic DNA. Polymerase ribozymes were prepared by in vitro transcription of dsDNA that was generated by either PCR amplification of the corresponding plasmid DNA or by PCR assembly of synthetic oligodeoxynucleotides. All RNA primers, templates, and ribozymes were purified by denaturing polyacrylamide gel electrophoresis (PAGE) and ethanol precipitation prior to use. His-tagged T7 polymerase was prepared from *Escherichia coli* strain BL21 containing plasmid pBH161 (kindly provided by W McAllister, SUNY Downstate Medical Center, Brooklyn, NY). Hot Start OneTaq was obtained from New England BioLabs (Ipswich, MA), rAPid alkaline phosphatase was from Sigma-Aldrich (St. Louis, MO), T4 polynucleotide kinase was from New England Biolabs (Ipswich, MA), and QIAprep Spin Miniprep Kit was from Qiagen (Germantown, MD). MyOne C1 streptavidin magnetic beads, PureLink PCR cleanup kit, TOPO TA cloning kit, SuperScript IV reverse transcriptase, Turbo DNase, and RNase T1 all were from Thermo Fisher Scientific (Waltham, MA). NTPs were from Chem-Impex International (Wood Dale, IL) and all other chemical reagents were from Sigma-Aldrich. The pH of Tris-HCl was adjusted at 23 °C.

### Assembly PCR
Polymerase ribozymes containing specific mutations were prepared by PCR assembly of synthetic oligodeoxynucleotides (*Supplementary file 4*), followed by in vitro transcription. The polymerase-encoding DNA was provided as six fragments, each overlapping by 20–22 base pairs. The fragments were assembled and amplified by PCR, using 0.5 µM each of the two outermost fragments and 0.005 µM each of the four internal fragments, and 0.025 U/µL OneTaq Hot Start polymerase, carried out for 25 thermal cycles. The PCR products were used directly in the in vitro transcription reaction.

### In vitro transcription
RNA templates and ribozymes were prepared by in vitro transcription in a mixture containing 5–20 ng/µL template DNA, 5 mM each NTP, 15 U/µL T7 RNA polymerase, 0.002 U/µL inorganic pyrophosphatase, 25 mM $MgCl_2$, 2 mM spermidine, 10 mM DTT, and 40 mM Tris-HCl (pH 8.0), which was incubated at 37 °C for 2 hr. The template DNA was then digested by adding 0.1 U/µL Turbo DNase and incubating at 37 °C for 1 hr.

### In vitro evolution
A starting pool of DNA templates was prepared by solid-phase synthesis, based on the sequence of the 38-6 polymerase and introducing random mutations at a frequency of 10 % per nucleotide for all positions between the two primer binding sites (nucleotides 15–167). The DNA was made double-stranded by primer extension using SuperScript IV reverse transcriptase, including 1.5 mM $MnCl_2$ in

the reaction mixture to promote extension through lesions that arose during DNA synthesis (**Chaput et al., 2003**). The dsDNA was amplified linearly by eight cycles of PCR using only the upstream primer, which introduced the T7 RNA polymerase promoter sequence. The DNA products were purified using the PureLink PCR cleanup kit, then 650 pmol dsDNA was used to prepare 2 nmol RNA to initiate the first round of evolution (round 39; see **Supplementary file 1**). The starting population consisted of an average of three copies each of $4 \times 10^{14}$ different RNAs. In all subsequent rounds, the size of the RNA population was 200 pmol.

In vitro evolution was carried out as described previously (**Tjhung et al., 2020**). The polymerase ribozymes were tethered at their 5' end to an RNA primer that was annealed to a complementary RNA template and extended by the ribozyme using the four NTPs. The resulting materials were subjected to the selection protocols described below, then reverse-transcribed, PCR-amplified, and forward-transcribed to yield progeny RNAs to begin the next round of evolution. Error-prone PCR (**Cadwell and Joyce, 1992**) was performed after rounds 43–50.

During rounds 39 and 40, the RNA template was biotinylated and all template-bound materials were captured on streptavidin-coated magnetic beads, which were washed twice with a solution of 8 M urea, 1 mM EDTA, 10 mM Tris-HCl (pH 8.0), and 0.05 % Tween-20. The extended products were then eluted from the template with a solution containing 25 mM NaOH, 1 mM EDTA, and 0.05 % Tween-20, neutralized with HCl, and precipitated with ethanol. The wash and elution conditions were optimized to exclude polymerases that failed to extend the attached primer, while retaining those that had extended the primer to yield full-length products.

In all subsequent rounds, selection was based on the ability of the polymerase to synthesize a functional hammerhead ribozyme. In those rounds, the 5' end of the polymerase was tethered to the 5' end of an RNA primer via a synthetic linker that contained both a biotin moiety and a substrate for the hammerhead ribozyme. The primer was then annealed through Watson-Crick pairing to a separate template encoding the sequence of the hammerhead. Following extension of the primer by the polymerase to generate the hammerhead ribozyme, the full-length products were purified by PAGE, then bound to streptavidin beads in the presence of 1 mM EDTA, 300 mM NaCl, 10 mM Tris-HCl (pH 8.0), and 0.05 % Tween-20, which prevented premature cleavage of the substrate by the hammerhead. The beads were washed with this same solution, then incubated in the presence of 20 mM MgCl$_2$ at 23 °C for 30 min. Under the latter conditions, active hammerhead ribozymes cleaved the attached RNA substrate, thereby releasing the corresponding polymerase from the beads.

Over the course of evolution, both the time allotted for RNA polymerization and the concentration of MgCl$_2$ were reduced (**Supplementary file 1**). Following round 52, the PCR-amplified DNA was cloned into *E. coli* using the TOPO-TA cloning kit, and the cells were grown at 37 °C for 16 hr on LB agar plates containing 50 µg/mL kanamycin. Individual colonies were picked and grown in 3 mL of LB medium with 50 µg/mL kanamycin at 37 °C for 16 hr. Plasmid DNA was harvested using the QIAprep Spin Miniprep Kit and sequenced by Eton Bioscience (San Diego, CA).

## RNA-catalyzed polymerization of RNA

RNA polymerization reactions used 100 nM ribozyme, 80 nM fluorescein- and biotin-labeled RNA primer, and 100 nM RNA template, which were annealed by heating at 80 °C for 30 s and then cooling to 17 °C. The annealed RNAs were added to a reaction mixture containing 4 mM each NTP, either 50 or 200 mM MgCl$_2$, 25 mM Tris-HCl (pH 8.3), and 0.05 % Tween-20, which was incubated at 17 °C. The reaction was quenched by manually adding an equal volume of a solution containing 250 mM EDTA (pH 8.0), 500 mM NaCl, 5 mM Tris-HCl (pH 8.0), and 0.025 % Tween-20, then mixed with 5 µg streptavidin magnetic beads per pmol biotinylated RNA primer, and incubated with gentle agitation at room temperature for 30 min. Prior to use, the beads had been washed according to the manufacturer's protocol, then incubated with 1 mg/mL tRNA in a solution containing 1 M NaCl, 1 mM EDTA, and 10 mM Tris-HCl (pH 8.0) for 30 min. The RNA template was removed from the bead-bound materials by two washes with a solution containing 25 mM NaOH, 1 mM EDTA, and 0.05 % Tween-20, followed by two washes with a solution containing 8 M urea, 1 mM EDTA, and 10 mM Tris-HCl (pH 8.0). Then the reaction products were eluted from the beads by incubating in 95 % formamide and 10 mM EDTA at 95 °C for 10 min, and were analyzed by PAGE. For fast-reaction kinetics, all reaction components other than the NTPs were pre-incubated at 17 °C for 5 min, then the NTPs were added and the solution was rapidly mixed to initiate the reaction.

## RNA-catalyzed ligation of RNA

The 52-2 polymerase was used to synthesize the class I ligase ribozyme by extending a 20-nucleotide RNA primer on a complementary RNA template. The resulting full-length products were purified by PAGE and subsequent ethanol precipitation. RNA ligation reactions were performed as described previously (*Tjhung et al., 2020*), using the same oligonucleotide substrates employed in previous kinetic studies (*Bergman et al., 2000*). The reaction mixture contained 20 μM 5'-substrate that had been fluorescently labeled with Cy5, 80 μM 3'-substrate that had been chemically triphosphorylated, either no or 1 μM ligase ribozyme, 60 mM MgCl$_2$, 200 mM KCl, 0.6 mM EDTA, and 50 mM Tris-HCl (pH 8.3), which was incubated at 23 °C for 24 hr. The reaction was quenched by adding four volumes of 95 % formamide and 20 mM EDTA and the products were analyzed by PAGE.

## Analysis of polymerase fidelity

The hammerhead and class I ligase ribozymes were synthesized by the 52-2 polymerase under standard reaction conditions. For the hammerhead, all partial- and full-length products, obtained in the presence of either 50 or 200 mM MgCl$_2$, were analyzed. For the ligase, only full-length products obtained in the presence of 200 mM MgCl$_2$ were analyzed. The products were converted to dsDNA molecules for Illumina sequencing, as described previously (*Tjhung et al., 2020*). Sequencing was carried out by the Salk Next Generation Sequencing Core on an Illumina MiniSeq, with either a 75- or 150-cycle paired-end run for the hammerhead or ligase, respectively.

The sequence data were processed to categorize all mutations relative to the expected sequence, as described previously (*Tjhung et al., 2020*). For both the hammerhead and ligase ribozyme, a custom JavaScript (source code in *Tjhung et al., 2020*) was used to calculate the number of matches, mismatches, deletions, and insertions as a function of template position and read length along the reference sequence. For the ligase, the distribution of Levenshtein distances from the reference sequence was determined directly from the alignment. The resulting data were manually processed to generate fidelity tables and position-specific data plots for the full-length products. HTS data, scripts, and related files are archived at the Dryad Digital Repository: https://doi.org/105061/dryadc866t1g78.

## In-line probing

The 5' end of the ribozyme was dephosphorylated using rAPid alkaline phosphatase, then [5'-$^{32}$ P]-labeled with [γ-$^{32}$P]ATP using T4 polynucleotide kinase, both according to the manufacturer's protocol. In-line probing (*Soukup and Breaker, 1999*) of 5'-labeled ribozymes was performed under the same conditions as the polymerization reaction, including the RNA primer, template, and four NTPs in the mixture. After 3, 6, 12, or 24 hr, the reaction was quenched with EDTA and the products were analyzed by PAGE. Individual bands in the gel were quantitated using ImageQuant 8.2. The raw counts were corrected by subtracting background counts, then scaled to the nucleotide position within the region of interest that had the highest level of cleavage.

## Analysis of the evolving population by deep sequencing

Sequencing of PCR products obtained after various rounds of evolution was performed at the Yale Center for Genome Analysis on an Illumina NovaSeq 6000, which generated ~20 million paired reads for each round that was sampled. The sequence datasets were quality-filtered, and trimmed using the paired-end read merger program PEAR (*Zhang et al., 2014*). The data were then filtered to include only reads of >150 nucleotides with a Phred score >33. Individual sequences were enumerated and converted to the fastq file format using a custom Python script (*Portillo et al., 2021*). The file sizes were reduced by removing sequences with <10 reads for rounds 16 and 31, <10,000 reads for round 27, and <1000 reads for all other rounds. The fastq file entries were then aligned using MUSCLE (*Edgar, 2004*). The aligned reads were trimmed to the region encompassing the P7 and P8 stems (nucleotides 9–17 and 83–95) using AliView (*Larsson, 2014*), then clustered using cd-hit-est (*Weizhong and Godzik, 2006*), with a clustering threshold of 100 % identity (-c 1.0), maximum unmatched length of two nucleotides (-U 2), and length difference cutoff of two nucleotides (-S 2). Clusters with >1% representation in any given round were identified. The insertion/deletion of A residues between nucleotides 17 and 18, and the presence of single mutations outside positions 11–16 and 84–89, were treated as representing the same cluster. HTS data, scripts, and related files are archived at the Dryad Digital Repository: https://doi.org/105061/dryadc866t1g78.

## Acknowledgements

The authors thank Adam Roth and Noah Setterholm for helpful discussions, and James Knight of the Yale Center for Genome Analysis for consultation on bioinformatics.

## Additional information

### Funding

| Funder | Grant reference number | Author |
| --- | --- | --- |
| National Aeronautics and Space Administration | NSSC19K0481 | Gerald F Joyce |
| Simons Foundation | 287624 | Gerald F Joyce |
| National Institutes of Health | P01GM022778 | Ronald R Breaker |
| Howard Hughes Medical Institute | | Ronald R Breaker |
| National Science Foundation | DGE1752134 | Xavier Portillo |

The funders had no role in study design, data collection and interpretation, or the decision to submit the work for publication.

### Author contributions

Xavier Portillo, Data curation, Formal analysis, Investigation, Methodology, Software, Validation, Visualization, Writing - review and editing; Yu-Ting Huang, Data curation, Formal analysis, Investigation, Methodology; Ronald R Breaker, Funding acquisition, Methodology, Project administration, Resources, Supervision, Writing - review and editing; David P Horning, Conceptualization, Data curation, Formal analysis, Investigation, Methodology, Project administration, Software, Supervision, Validation, Visualization, Writing - original draft, Writing - review and editing; Gerald F Joyce, Conceptualization, Data curation, Formal analysis, Funding acquisition, Investigation, Methodology, Project administration, Resources, Supervision, Validation, Visualization, Writing - original draft, Writing - review and editing

### Author ORCIDs

Xavier Portillo (i) http://orcid.org/0000-0003-1225-1133
Ronald R Breaker (i) http://orcid.org/0000-0002-2165-536X
David P Horning (i) http://orcid.org/0000-0003-3357-6092
Gerald F Joyce (i) http://orcid.org/0000-0003-0603-2874

### Decision letter and Author response

Decision letter https://doi.org/10.7554/eLife.71557.sa1
Author response https://doi.org/10.7554/eLife.71557.sa2

## Additional files

### Supplementary files

• Supplementary file 1. Parameters for directed evolution of polymerase ribozymes. The third column indicates the number of nucleoside 5'-triphosphates (NTPs) to be added in order to meet the selection criterion. '+' indicates PCR mutagenesis; '10%' indicates random mutagenesis of the 38-6 polymerase at 10 % degeneracy per position.

• Supplementary file 2. Fidelity of the 52-2 polymerase. The 52-2 polymerase was used to synthesize the hammerhead and class I ligase ribozymes (see *Supplementary file 4* for sequences). The hammerhead was synthesized in the presence of either 200 or 50 mM $Mg^{2+}$ and the ligase was synthesized in the presence of 200 mM $Mg^{2+}$. The full-length products were analyzed by deep sequencing to obtain the frequencies of each type of mutation. The average fidelity was calculated as the geometric mean of the fidelities for each templating nucleobase, which gave values of 91.7 %

or 94.4 % for the hammerhead with either 200 or 50 mM Mg$^{2+}$, respectively, and 84.1 % for the ligase. Deletions were included as mutations and insertions were treated as a single mutation at the immediately upstream position.

• Supplementary file 3. Prevalent sequence clusters over the course of evolution. Clusters representing >1% of the population in a given round are shown. Clusters that differ by only a single nucleotide outside the P8 stem are given the same name. Clusters that include the wild type, 24-3, 38-6, or 52-2 polymerase are given the name of that polymerase.

• Supplementary file 4. Sequences of RNA and DNA molecules used in this study. The molecules were synthesized in-house (syn), purchased from IDT (com), or prepared by in vitro transcription (ivt). The T7 RNA polymerase promoter sequence is underlined. Sequences in green indicate the complementary tag on the ribozyme and templates used to improve processivity. Sequences in blue indicate the primer binding site for RNA-templated RNA polymerization. Nucleotides in red are mutations relative to the 52-2 polymerase, that were introduced during PCR assembly. hex, hexynyl group for click chemistry; PEG, polyethylene glycol linker; pcl, photocleavable linker; FAM, 6-fluorescein label; Cy5, cyanine 5-methine label; ppp, 5'-triphosphate added by chemical synthesis.

• Transparent reporting form

• Source data 1. Cluster analysis of the nucleotide sequence at positions 9–17 and 83–95 of the polymerase over the course of evolution.

• Source code 1. Python script used to identify unique polymerase sequences and their corresponding number of reads for each round of evolution that was analyzed.

### Data availability

Source data, including raw gel electrophoresis images (*.tiff files) and tables listing all datapoints and model fitting parameters, for all applicable graphs and plots, are provided for Figures 1, 2-supplement-2, 3 (and supplements 1 and 2), 4 (and supplement 1), and 5 (and supplement 5). High-throughput sequencing data and analysis used to generate Figure 1-supplement 1, Figure 5, and Supplementary file 2 are available and archived at the Dryad Digital Repository.

The following dataset was generated:

| Author(s) | Year | Dataset title | Dataset URL | Database and Identifier |
|---|---|---|---|---|
| Portillo X, Huang Y, Breaker R, David H, Joyce G | 2021 | Polymerase fidelity data from: Witnessing the structural evolution of an RNA enzyme | https://doi.org/10.5061/dryad.c866t1g78 | Dryad Digital Repository, 10.5061/dryad.c866t1g78 |

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
