## [Decision Letter]

**Acceptance summary:**

Thank you for the thorough and thoughtful revision of your manuscript. We think that it is a very important contribution with regard to the evolution of catalytic RNAs and the origin of life.

**Decision letter after peer review:**

Thank you for submitting your article "Witnessing the structural evolution of an RNA enzyme" for consideration by *eLife*. Your article has been reviewed by 3 peer reviewers, and the evaluation has been overseen by Timothy Nilsen as Reviewing Editor and James Manley as the Senior Editor. The following individuals involved in review of your submission have agreed to reveal their identity: Peter J Unrau (Reviewer #1); Donald Burke (Reviewer #3).

Essential revisions:

All of the reviewers found the work to be quite interesting and important to the general field of RNA catalysis. Nevertheless, each reviewer made a number of suggestions for clarifications that could be handled by text revision. In addition, some new experiments thought to be straightforward were also suggested. While there is no requirement to perform these experiments, it was the consensus that they would strengthen the paper. We reiterate that it is your option to either perform the experiments or not. When revision is complete please resubmit together with a letter detailing all changes and your response to reviewers' comments.

*Reviewer #1 (Recommendations for the authors):*

I have a two suggestions for this manuscript, which was a pleasure to read:

1. Figure 5 summarizes a lot of high detailed sequence information centered on understanding the emergence of the new P8 helix. This figure shows highly represented sequence clusters and their sequential emergence and decline through the evolutionary process. Each cluster results from mutations in the top and the bottom strands of the P8 helix, but are such mutations correlated across clusters when considered by strand? So for example does the second G in the top strand of 24-3 result in the G in 18.1, 24.1, 31.2 and 31.3? Similarly on the bottom strand do the double U mutations in 11.3 give rise to the same mutations in 38.6 and 31.1? It would seem that for each cluster one could quite easily plot the frequency of the top strand and the bottom strands as a function of round (say show the top strand frequencies in shades of green, the bottom strand frequencies in shades of red and the intersection in shaded of blue as currently shown). This could reveal some interesting information about the evolutionary progression that is currently suppressed in the figure. So perhaps the 24-3 top strand pattern declines when 31.3 top strand mutation emerges etc?

2. Further to this point: Are there evolutionary correlations between the emergence of the new P8 helix and the additional mutations found in the accessory domain of the polymerase? Figure 2 (Supplement 1) seems to suggest this possibility. One could simply hypothesize that the new P8 structure rotates the P7-P6-P3 stack and requires compensatory mutations in other regions of the ribozyme as a result. I was therefore a bit puzzled as to why the authors do not search for such a potential correlation. It would be a fairly simple analysis to explore and would place the importance of the P8 helical mutations in a clearer evolutionary perspective.

*Reviewer #2 (Recommendations for the authors):*

1) "…. provide conditions that are conducive to both increased polymerase fidelity and reduced degradation of RNA." How is low magnesium concentration conducive to increased polymerase fidelity? A comment or reference for this statement should be included.

2) The paper analyzes the endpoint of 52 consecutive rounds of directed evolution under progressively more demanding selection conditions. However, it's important for the authors to better clarify from the start that this manuscript consists of only 14 new rounds of selection (and a deeper analysis of rounds of selection previously carried out and described in earlier publications).

In the same vein, the authors should define the "wild-type polymerase" as the starting point of their directed evolution experiments leading to 24-3 (PNAS 113: 9786) and finally to 52-2 described here.

3) The template 5 ´-GUGUGGAGUGACCUCUCCUGUGUGAGUG-3 ´ doesn't seem to be defined nor described in the oligo table.

4) Given the substantial comparisons between older generations of ribozymes, it would be useful for the reader to have a supplementary figure containing the wildtype sequence with the same notation and numbering showed in figure 2 for 52-2.

5) P10 l6 refer to figure 5 not 4.

*Reviewer #3 (Recommendations for the authors):*

Abstract, last line: Change "suggesting" to "raising the possibility," "hence, we suggest," or some similar phrase that does not imply a direct extrapolation from the data in this paper to predictions of how future rounds of evolution may play out.

The last sentence of p6 states that the new stem (P8) is "mutually exclusive to the prior core structure of the ribozyme." This is overstated. P7 forms in both the 'old' and 'new' structures, although it is shorter in the 'new' structure because some of the nucleotides have shifted pairing partners to form P8.

At the bottom of p9, please give an indication of the sizes of the datasets that went into RNASeq analyses, as this speaks to the relative strength of the analysis. Materials and methods notes that the NovaSeq6000 can generate ~20M reads per sample, but there is no indication as to whether "sample" refers to the entire sample to be sequenced (e.g., all pools labeled with indices and merged) or to each population separately. My guess is that there were ~1M reads per population for the 19 sequenced populations; please clarify.

On p16, this sentence is unclear: "In those rounds, the polymerase was tethered to an RNA primer that was linked to a biotinylated substrate for the hammerhead ribozyme." Presumably the primer 3' OH was free to be extended; hence, both the 'tethering' (to the polymerase) and 'linking' (to biotinylated substrate) must occur elsewhere. Is this a branched molecule? Please clarify the molecular nature of this part of the experimental design.

p5, middle. Change "A starting population … were used" to "A starting population … was used." Later on same page: Change "A total … were carried out" to "A total … was carried out." (The subjects of both sentences are singular.) Alternatively, one or both sentences could be reworded, such as, "Fourteen rounds of evolution were carried out."

p5, bottom, please specify how many individuals were cloned and sequenced.

p8, third line from bottom, omit "relatively."

p10, line 6, change "Figure 4" to "Figure 5."

p15, line 6. Change "A starting pool … were prepared" to "A starting pool … was prepared."

p16, lines 7-9. Please clarify these points as related to the hammerhead ribozyme, to the extent that each is applicable. 1) Change "The primer was then bound" to "The primer was then annealed" (if the nature of the 'binding' is Watson-Crick base pairing). 2) Change beginning of next sentence to "Following extension of the primer by the polymerase to produce full-length hammerhead ribozyme, the full-length products were …" 3) Note presence of EDTA during library capture onto StrAv beads to prevent premature hammerhead ribozyme cleavage.

On pp16-17, please specify whether the fast kinetic reactions were quenched manually (which appears to be the case from the overall description).

---

## [Author Response]

Essential revisions:All of the reviewers found the work to be quite interesting and important to the general field of RNA catalysis. Nevertheless, each reviewer made a number of suggestions for clarifications that could be handled by text revision. In addition, some new experiments thought to be straightforward were also suggested. While there is no requirement to perform these experiments, it was the consensus that they would strengthen the paper. We reiterate that it is your option to either perform the experiments or not. When revision is complete please resubmit together with a letter detailing all changes and your response to reviewers' comments.Reviewer #1 (Recommendations for the authors):I have a two suggestions for this manuscript, which was a pleasure to read:1. Figure 5 summarizes a lot of high detailed sequence information centered on understanding the emergence of the new P8 helix. This figure shows highly represented sequence clusters and their sequential emergence and decline through the evolutionary process. Each cluster results from mutations in the top and the bottom strands of the P8 helix, but are such mutations correlated across clusters when considered by strand? So for example does the second G in the top strand of 24-3 result in the G in 18.1, 24.1, 31.2 and 31.3? Similarly on the bottom strand do the double U mutations in 11.3 give rise to the same mutations in 38.6 and 31.1? It would seem that for each cluster one could quite easily plot the frequency of the top strand and the bottom strands as a function of round (say show the top strand frequencies in shades of green, the bottom strand frequencies in shades of red and the intersection in shaded of blue as currently shown). This could reveal some interesting information about the evolutionary progression that is currently suppressed in the figure. So perhaps the 24-3 top strand pattern declines when 31.3 top strand mutation emerges etc?

We analyzed the frequency of mutations for both the top and bottom strands of the new P8 stem over the course of evolution, considering each strand individually. For each of the major forms of the polymerase, there is a high degree of correlation between the frequency of occurrence of particular variants of the two strands, indicating that they rose and fell in concert. These data are shown in a new figure supplement to Figure 5 and are discussed in the text.

2. Further to this point: Are there evolutionary correlations between the emergence of the new P8 helix and the additional mutations found in the accessory domain of the polymerase? Figure 2 (Supplement 1) seems to suggest this possibility. One could simply hypothesize that the new P8 structure rotates the P7-P6-P3 stack and requires compensatory mutations in other regions of the ribozyme as a result. I was therefore a bit puzzled as to why the authors do not search for such a potential correlation. It would be a fairly simple analysis to explore and would place the importance of the P8 helical mutations in a clearer evolutionary perspective.

No strong correlations were observed between specific mutations within the pseudoknot region and those in the 3´ accessory domain. To investigate this issue biochemically, and prompted by the suggestion of Reviewer 3, we “transplanted” the new pseudoknot structure onto the wild-type polymerase, without any of the other mutations, and found that the resulting chimeric molecule has comparable activity to that of the final evolved 52-2 polymerase. These results are described in the text. We revised Figure 2—figure supplement 1 to include the sequence of the chimeric construct and added figure supplement 2 to show the comparative activity of the wild-type, 52-2, and chimeric polymerases.

Reviewer #2 (Recommendations for the authors):1) "…. provide conditions that are conducive to both increased polymerase fidelity and reduced degradation of RNA." How is low magnesium concentration conducive to increased polymerase fidelity? A comment or reference for this statement should be included.

We provided two references describing how lowering the Mg^2+^ concentration can reduce the frequency of mutation by a polymerase. Also, in response to point #2, we carried out deep sequencing analysis of the ribozyme-synthesized products, generated in the presence of either 50 or 200 mM MgCl2. We did this for the hammerhead ribozyme, which is obtained in reasonable yield at either concentration, and found the error rate is indeed lower at 50 compared to 200 mM MgCl2 (5.6% vs. 8.3%, respectively). We added a discussion of polymerase fidelity as a function of both template length and Mg^2+^ concentration.

2) The paper analyzes the endpoint of 52 consecutive rounds of directed evolution under progressively more demanding selection conditions. However, it's important for the authors to better clarify from the start that this manuscript consists of only 14 new rounds of selection (and a deeper analysis of rounds of selection previously carried out and described in earlier publications).In the same vein, the authors should define the "wild-type polymerase" as the starting point of their directed evolution experiments leading to 24-3 (PNAS 113: 9786) and finally to 52-2 described here.

We added a sentence to make clear that rounds 1–38 of directed evolution were carried out in our previous studies and that only rounds 39–52 are new here. However, the present study takes into account the entire lineage to demonstrate the emergence of the novel structure. We explain that the “wild type” of the present study is the same construct that was used to launch the first round of evolution.

3) The template 5 ´-GUGUGGAGUGACCUCUCCUGUGUGAGUG-3 ´ doesn't seem to be defined nor described in the oligo table.

Supplementary file 3. We are grateful to the reviewer for catching the transposition error involving the sequence of template “T1”. There was a similar error involving template “R8”. Both errors were corrected.

4) Given the substantial comparisons between older generations of ribozymes, it would be useful for the reader to have a supplementary figure containing the wildtype sequence with the same notation and numbering showed in figure 2 for 52-2.

We added a figure supplement to Figure 2 showing the numbered sequence and secondary structure of the wild-type polymerase. This added figure also gave us the opportunity to depict the chimeric molecule in which the pseudoknot was imported onto the wild-type background.

5) P10 l6 refer to figure 5 not 4.

This typo was corrected.

Reviewer #3 (Recommendations for the authors):Abstract, last line: Change "suggesting" to "raising the possibility," "hence, we suggest," or some similar phrase that does not imply a direct extrapolation from the data in this paper to predictions of how future rounds of evolution may play out.

Abstract, last line: The wording was changed as suggested.

The last sentence of p6 states that the new stem (P8) is "mutually exclusive to the prior core structure of the ribozyme." This is overstated. P7 forms in both the 'old' and 'new' structures, although it is shorter in the 'new' structure because some of the nucleotides have shifted pairing partners to form P8.

We clarified this statement to explain that only the distal portion of the P7 stem has been repurposed to form the new pseudoknot and that the proximal portion of this stem remains intact.

At the bottom of p9, please give an indication of the sizes of the datasets that went into RNASeq analyses, as this speaks to the relative strength of the analysis. Materials and methods notes that the NovaSeq6000 can generate ~20M reads per sample, but there is no indication as to whether "sample" refers to the entire sample to be sequenced (e.g., all pools labeled with indices and merged) or to each population separately. My guess is that there were ~1M reads per population for the 19 sequenced populations; please clarify.

We now explain that in carrying out Illumina sequencing we obtained ~20 million paired reads for *each* of the 19 different rounds that were sampled.

On p16, this sentence is unclear: "In those rounds, the polymerase was tethered to an RNA primer that was linked to a biotinylated substrate for the hammerhead ribozyme." Presumably the primer 3' OH was free to be extended; hence, both the 'tethering' (to the polymerase) and 'linking' (to biotinylated substrate) must occur elsewhere. Is this a branched molecule? Please clarify the molecular nature of this part of the experimental design.

We clarified the description of how the polymerase was linked both to biotin and to the substrate for the hammerhead ribozyme.

p5, middle. Change "A starting population … were used" to "A starting population … was used." Later on same page: Change "A total … were carried out" to "A total … was carried out." (The subjects of both sentences are singular.) Alternatively, one or both sentences could be reworded, such as, "Fourteen rounds of evolution were carried out."

The wording was changed as suggested.

p5, bottom, please specify how many individuals were cloned and sequenced.

We now explain that 30 individuals were cloned and sequenced.

p8, third line from bottom, omit "relatively."

The wording was changed as suggested.

p10, line 6, change "Figure 4" to "Figure 5."

This typo was corrected.

p15, line 6. Change "A starting pool … were prepared" to "A starting pool … was prepared."

The wording was changed as suggested.

p16, lines 7-9. Please clarify these points as related to the hammerhead ribozyme, to the extent that each is applicable. 1) Change "The primer was then bound" to "The primer was then annealed" (if the nature of the 'binding' is Watson-Crick base pairing). 2) Change beginning of next sentence to "Following extension of the primer by the polymerase to produce full-length hammerhead ribozyme, the full-length products were …" 3) Note presence of EDTA during library capture onto StrAv beads to prevent premature hammerhead ribozyme cleavage.

The wording was changed as suggested, and we now explain how EDTA was used to prevent the hammerhead ribozyme from cleaving the substrate prematurely.

On pp16-17, please specify whether the fast kinetic reactions were quenched manually (which appears to be the case from the overall description).

Indeed the reactions were quenched manually, as we now describe.